# Plasma Redox Balance in Advanced-Maternal-Age Pregnant Women and Effects of Plasma on Umbilical Cord Mesenchymal Stem Cells

**DOI:** 10.3390/ijms25094869

**Published:** 2024-04-29

**Authors:** Elena Grossini, Carmen Imma Aquino, Sakthipriyan Venkatesan, Libera Troìa, Eleonora Tizzoni, Federica Fumagalli, Daniela Ferrante, Rosanna Vaschetto, Valentino Remorgida, Daniela Surico

**Affiliations:** 1Laboratory of Physiology, Department of Translational Medicine, Università del Piemonte Orientale, Via Solaroli 17, 28100 Novara, Italy; sakthipriyan.venkatesan@uniupo.it; 2Gynecology and Obstetrics, Department of Translational Medicine, Università del Piemonte Orientale, “Maggiore della Carità” Hospital, 28100 Novara, Italy; 20033548@studenti.uniupo.it (C.I.A.); 20021710@studenti.uniupo.it (E.T.); 20020579@studenti.uniupo.it (F.F.); valentino.remorgida@med.uniupo.it (V.R.); daniela.surico@med.uniupo.it (D.S.); 3Medical Statistics, Department of Translational Medicine, Università del Piemonte Orientale, 28100 Novara, Italy; daniela.ferrante@med.uniupo.it; 4Anesthesia and Intensive Care, Department of Translational Medicine, Università del Piemonte Orientale, 28100 Novara, Italy; rosanna.vaschetto@med.uniupo.it

**Keywords:** aging, antioxidants, lipid peroxidation, nitric oxide, pregnancy

## Abstract

Pregnancy at advanced maternal age (AMA) is a condition of potential risk for the development of maternal–fetal complications with possible repercussions even in the long term. Here, we analyzed the changes in plasma redox balance and the effects of plasma on human umbilical cord mesenchymal cells (hUMSCs) in AMA pregnant women (patients) at various timings of pregnancy. One hundred patients and twenty pregnant women younger than 40 years (controls) were recruited and evaluated at various timings during pregnancy until after delivery. Plasma samples were used to measure the thiobarbituric acid reactive substances (TBARS), glutathione and nitric oxide (NO). In addition, plasma was used to stimulate the hUMSCs, which were tested for cell viability, reactive oxygen species (ROS) and NO release. The obtained results showed that, throughout pregnancy until after delivery in patients, the levels of plasma glutathione and NO were lower than those of controls, while those of TBARS were higher. Moreover, plasma of patients reduced cell viability and NO release, and increased ROS release in hUMSCs. Our results highlighted alterations in the redox balance and the presence of potentially harmful circulating factors in plasma of patients. They could have clinical relevance for the prevention of complications related to AMA pregnancy.

## 1. Introduction

The historical definition of the term “Advanced Maternal Age” (AMA) refers to a condition of maternal age over 35 years [1,2]. The analysis of the data collected in the last three decades evidenced an increase in the average age of conception and delivery in women belonging to developed countries [3]. These findings concern both women aged 35–39 years, in whom the increase in the birth rate was from 45.9 per 1000 women in 2010 to 52.7 per 1000 women in 2019, and women aged 40–44 years, who showed an increase in the birth rate from 10.2 to 12 per 1000 women in the same range of years [4,5,6]. In addition to reasons related to work and career needs, the increase in AMA pregnant women could be explained also by the development in advanced reproductive technology, which has extended the reproductive window [7,8]. Thus, on the grounds of these considerations, the above reported definition of AMA appears outdated and it would be more appropriate to use it for pregnant women older than 40 years [9,10].

It should be kept in mind that AMA pregnancy could represent a risk factor for adverse maternal complications, such as pre-eclampsia, gestational diabetes mellitus, gestational hypertension, and Cesarean delivery, as well as for fetal outcomes [7,8]. In addition to increased maternal mortality and morbidity, the above health complications also have repercussions in terms of increased costs to health systems worldwide [11]; a great deal of effort should be made to try to counteract or even prevent them.

Actions aimed at preventing AMA pregnancy-related complications could be focused on the modulation of its pathophysiological mechanisms.

In this regard, an important role is played by the loss of placental functions [12]. Indeed, changes in the placentation process may have serious consequences in pregnancy for women presenting specific characteristics, including AMA, which are also related to endothelial dysfunction [11].

In physiologic pregnancy, reactive oxygen species (ROS) and reactive nitrogen species (RNS) are generated mainly by the placenta and vascular endothelium, as well as through the action of xanthine oxidase and nitric oxide (NO) synthase [13,14].

However, when the feto-placental unit is poorly perfused, such as during AMA pregnancy, an increased oxidant release not counteracted by antioxidants like superoxide dismutase or glutathione (GSH) and glutathione peroxidase may affect normal vasodilation and become an important factor in the pathogenesis of AMA pregnancy-related complications, like those affecting the cardiovascular system, kidney and liver as well [11,13,15].

In particular, increased concentrations of lipid peroxides in the villous and decidual tissues have been reported to be common to all miscarriages and have been correlated with proteinuria, uricemia and pulsatility index of umbilical arteries in pre-eclamptic pregnant women [16].

Furthermore, changes in lipid peroxides could play an important role in AMA pregnancy since they can affect endothelial function through changes in thromboxane A2 and NO release in the utero-placental and maternal systemic vasculature as well [17].

Also, as regarding GSH, which is one of the most important antioxidants that can reduce unstable ROS and turn into oxidized GSH, it was reported to be strongly reduced in the plasma of pre-eclamptic women [11,16], in placental aging and in adverse pregnancy outcomes [18].

It is important to underline that oxidative stress resulting from unbalance between oxidants and antioxidants can be involved in the physiopathology of AMA pregnancy through changes in endothelial NO availability and endothelial dysfunction.

Another possible target of the prevention/modulation of AMA pregnancy-related complications could be represented by the human umbilical cord mesenchymal stem cells (hUMSCs), which can protect placental function by increasing its development and angiogenesis through paracrine actions and the release of immunomodulatory factors [16,19,20,21]. Also, those cells can be involved in the modulation of oxidative stress/inflammation, have the ability to recover ovarian function in premature ovarian insufficiency or natural aging animal models, and exert cardioprotective effects through the release of microRNA [22].

Despite existing knowledge about this issue, however, additional information regarding the plasma redox balance and the changes in hUMSCs function induced by circulating factors in AMA pregnant women at different times of pregnancy would be mandatory to better understand the pathophysiological aspects.

Considering what has been reported above regarding the role of lipid peroxides, GSH and NO in the physiopathology of AMA pregnancy and the lack of detailed information, here, our primary endpoint was to examine the plasma levels of lipid markers of peroxidation, GSH and NO at the first trimester (T0), at the second trimester during the morphologic ultrasound (T1) and at 48–72 h after the delivery (T2) in pregnant women over 40 years (median age 41) compared with pregnant women younger than 40 years (median age 30). In particular, we used the thiobarbituric acid reactive substances (TBARS) assay for measurement of plasma lipid peroxides such as malonyldialdeide (MDA) and specific assays for GSH and NO, as previously conducted [23,24,25]. Secondly, we wanted to analyze the effects of plasma taken from AMA pregnant women at different time points on hUMSCs in order to highlight the role of circulating factors in the modulation of their viability and oxidant release.

## 2. Results

### 2.1. Patients

All patients were subjected to careful anamnestic investigation. Risk factors were also investigated for pregnancy disorders and previous pathological pregnancies (Table 1). The anamnestic characteristics of the two groups, compared by means of Fisher’s test, were found to be superimposable, with no statistically significant differences (*p* > 0.05) except for maternal age (Table 1). Therefore, the only characteristic that significantly differentiated the group of AMA patients and the group of controls was obviously the age (41 versus 30 years, *p* < 0.0001).

Although the body mass index (BMI) before pregnancy of AMA patients was higher than that of controls (*p* < 0.05), it was within the normal weight range.

A positive first trimester screening was found in 26 of 100 AMA pregnant patients, while it was positive in only one patient of the controls. In the calculation of the risk, the age of the patients had a substantial influence.

In Table 2, we describe the main delivery outcomes of patients and controls. The statistical analysis did not show a significant high percentage of complications in AMA patients vs. controls (*p* = 0.12). Among AMA patients, spontaneous delivery occurred in 23 patients (43.4%), Cesarean section in 22 (41.5%), induced childbirths in 5 (9.4%) and dystocic deliveries in 3 (5.7%) (Table 2). In controls, 11 (61.1%) spontaneous delivery occurred, childbirths were induced 3 times (16.7%), as Cesarean section, and dystocic delivery occurred only once (5.6%). A total of 28 patients presented no complications and 36 developed some gestational comorbidities, such as gestational diabetes, preterm birth, intrahepatic cholestasis of pregnancy, gestational hypertension, pre-eclampsia, placenta low lying, intrauterine growth restriction (IUGR) and deep vein thrombosis.

### 2.2. Plasma TBARS, GSH and NO

The results of plasma TBARS, GSH and NO measurements performed in AMA patients and controls at various times during pregnancy up to postpartum are shown in Table 3. All models were adjusted by type of delivery (classified as in Table 2).

It is of note that the overall mean of the considered outcomes (TBARS, GSH and NO) between AMA patients and controls was significantly different (group effect).

It is also of note that, at T3, in AMA patients, the levels of TBARS remained higher than those measured in controls (*p* < 0.05): in AMA patients, 9 πM 5.3–12 and, in controls, 5.1 πM (4.6–5.7). Also, plasma GSH of AMA patients was lower than that found in controls (*p* < 0.05): in AMA patients, 5 πM (3.4–7.2) and, in controls, 8 πM (7–8.9). As regards NO, in AMA patients, the plasma values were lower than those of controls (*p* < 0.05): in AMA patients, 9.6 πM (5.6–13) and, in controls, 14.5 πM (11.9–16.7).

Also, in the plasma of the umbilical cords, there was an alteration in the redox state in AMA patients. Indeed, GSH levels were lower than those observed in the controls (*p* < 0.05; Figure 1A), while TBARS levels were higher (*p* < 0.05; Figure 1B). Instead, we did not detect any significant differences regarding NO (Figure 1C).

### 2.3. Effects of Plasma on hUMSCs

The in vitro experiments showed that the treatment of hUMSCs with plasma taken from AMA patients was able to reduce cell viability and NO release and increase ROS production (Figure 2A–C). As evidenced in Figure 2A, the cell viability was lower than that measured both in hUMSCs treated with plasma of the controls and in the untreated hUMSCs.

If we analyze the results obtained at the three different time points, it can be observed that there are no significant variations in the median values of cell viability in hUMSCs treated with plasma of AMA patients (Figure 2A). In the case of the controls, cell viability was even increased at T3 in comparison with that found in the untreated hUMSCs (*p* < 0.05).

As performed for plasma measurements of TBARS, GSH and NO, we executed a statistical analysis about response trends over time in measured variables of the in vitro experiments too. As shown in Table 4, it is of note that a significant interaction group × time emerged for in vitro data about NO and cell viability. This means that changes in response over time differed among AMA patients and controls.

Concerning ROS, we observed an increased release by hUMSCs treated with plasma of AMA patients both vs. the untreated cells and vs. hUMSCs treated with plasma of the controls (Figure 2B).

Also, in hUMSCs treated with plasma of AMA patients at T3, cell viability and NO release were lower than those found in hUMSCs treated with plasma of controls, whereas ROS release was higher (Figure 2).

### 2.4. Clinical Pattern

The analysis of the data regarding AMA patients suffering from gestational diabetes, preterm birth, intrahepatic gestational cholestasis, gestational hypertension, pre-eclampsia, low-lying placenta, IUGR and deep vein thrombosis vs. controls, and even comparing patients with pregestational diseases vs. uncomplicated ones, did not show any statistically significant difference. Thus, the altered redox balance we found in AMA patients can be linked to age rather than to any comorbidities.

## 3. Discussion

Our results highlighted an altered redox balance in the plasma of AMA patients at diagnosis and throughout the duration of pregnancy up to 48 h postpartum. Moreover, we showed that unknown factors capable of affecting viability and ROS release in hUMSCs circulate in the plasma of the aforementioned patients.

In recent years, there has been a significant increase in AMA pregnancies, probably due to epidemiological and socio-economic reasons, as well as a progressive strengthening of medically assisted fertility techniques [2]. The percentage of women who decide to postpone pregnancy in developing countries is increasing. According to the recent Certificate of Assistance in Childbirth (CeDAP), the average age of mothers is 33.1 years for the Italian population, and the average age of having a first child is over 31 years, according to the global report [26,27].

In the past, an AMA gestation was almost always accidental and related to pluriparity; however, today it is more frequently associated with the first pregnancy and with changes in demographics of childbirth.

As previously reported, AMA pregnancy has been considered as an independent risk factor for several complications, varying across ages. In fact, these pregnancies are known to be linked to several comorbidities and adverse outcomes, such as pregnancy-related hypertensive disorders, blood transfusion, and maternal and fetal mortality [16,28]. In this regard, AMA has been reported to increase the risk of stillbirth (RR 2.16, 95% CI 1.86–2.51), perinatal mortality, intrauterine growth restriction, neonatal death, admission to neonatal intensive care unit, pre-eclampsia, preterm delivery, cesarean delivery and maternal mortality compared with women younger than 40 years old (RR 3.18, 95% CI 1.68–5.98) [29].

The relationship between AMA and stillbirth is not directly linked to maternal morbidity or assisted reproductive technology [30].

One of the possible explanations of those phenomena could be the increased oxidative stress. Pregnancy itself augments susceptibility to oxidative stress, with an elevated presence of ROS and RNS in the circulatory system. The amplified gestational redox imbalance can trigger several adverse outcomes such as abnormal placental function, and complications, including the forementioned pre-eclampsia, embryonic resorption, recurrent pregnancy loss, fetal developmental anomalies, intrauterine growth restriction and, in extreme instances, fetal death. The body’s answer to control the increase in RNS/ROS levels requires nonenzymatic and enzymatic defense processes (i.e., acting on TBARS, GSH and NO). The literature describes a plausible association between compromised antioxidant enzyme function and the occurrence of these adverse gestational outcomes. Oxidative stress causes detrimental consequences on maternal physiology, pregnancy progression and fetal development by acting on placental function and compromising oxygen and nutrient delivery to the fetus [31].

Pregnancies complicated by conditions such as oxidative stress, hypoxia and inflammation are associated with alterations in placental vasculogenesis, trophoblast expression of transporters and hormone production contributing to alteration in fetal development [32].

Regarding this issue, it should be pointed out that, in physiological pregnancy, the increase in oxidants is counterbalanced by the production of antioxidants and the occurrence of oxidative stress is limited [33,34]. In particular, it was shown that there is a gradual reduction in lipoperoxide formation with the progress of gestation to protect the fetus against ROS [35].

Instead, in all conditions in which the systems for regulating the balance between oxidants and antioxidants are poorly effective, the development of oxidative stress is markedly greater. Certainly, one of those conditions predisposing the onset of oxidative stress is aging [36].

The primary endpoint of this study was the analysis of plasma redox balance at different time points during pregnancy in AMA vs. younger pregnant women.

We measured lipid peroxidation as MDA release by means of the TBARS assay, since it has been widely used in previous studies aimed at analyzing oxidants in biologic fluids during pregnancy too [34,37].

We found that patients and controls differed in terms of plasma TBARS and GSH levels throughout pregnancy. Hence, lipid peroxidation markers were higher in women with advanced age, while the antioxidant was higher in the control. Similar results were noted also in the experiments performed by using plasma from umbilical cords of AMA patients. Although both age and body mass index were different among groups, mean body mass index for AMA women was within the normal weight range, suggesting differences in redox potential could be attributable to age [38].

TBARS are the most reliable biomarkers of oxidative stress [39] and have been shown to have some clinical relevance in aging and age-related pathologies [40]. Our data about TBARS are in agreement with those found by de Lucca L et al. and Draganovic D et al. in complicated and uncomplicated pregnancies and in pregnancy-induced hypertension women, respectively [32,35]. Indeed, among the 45 pregnant women recruited in the study by de Lucca L et al., TBARS increased in the second trimester when compared to the first trimester of pregnancy in the uncomplicated group, whereas they increased in the third trimester when compared to the first trimester of pregnancy in the complicated group. Also, the 100 pregnant women with hypertension recruited by Draganovic D et al. had increased mean TBARS values over 20 µmol, whereas, in healthy pregnant women, only 1% experienced increased TBARS value.

Regarding GSH, it was suggested that the measurement of its plasma levels could provide a way to measure resilience of redox networks in aging and age-related disease [41], and it is strongly reduced in placental aging and in adverse pregnancy outcomes [18]. For this reason, we have performed GSH measurements in plasma of AMA patients throughout pregnancy and compared it with those of controls.

Also, although a role of the above reported pregnancy-related complications in the alteration of the balance between oxidants and antioxidants is conceivable [42,43], the statistical analysis showed that there were no significant differences regarding the parameters of redox balance and oxidative stress between AMA patients with complications and without complications. In fact, 28 patients presented no complications and 36 developed some gestational comorbidities. Indeed, as described in the Results section, the analysis of the data regarding the 36 AMA patients suffering from pathologies during pregnancy vs. controls, and even comparing patients with pregestational diseases vs. uncomplicated ones, did not show any statistically significant difference.

For this reason, we can attribute the increased values of plasma TBARS and reduced values of plasma GSH more to the advanced age of the patients than to their complications.

The data we obtained about the redox balance in AMA pregnant women up to 48 h after delivery and in umbilical blood could have clinical implications in those subjects and newborns. In this regard, there are data demonstrating that both children born from complicated pregnancies and AMA pregnant women are at increased risk of cardiovascular disease, which could be related to oxidative stress and endothelial dysfunction that had arisen during pregnancy [42,43,44,45,46,47].

In particular, our results could be helpful to increase knowledge about the role of oxidative stress in the physiopathology of AMA pregnancy and of systemic diseases that could arise in AMA pregnant women [44,45]. Also, it could stimulate studies on the efficacy of preventive or therapeutic treatments aimed at contrasting them.

As reported above, a key role as a pathogenic mechanism could be played by oxidative stress and endothelial dysfunction [12,34,46,47,48]. The data we obtained about plasma levels of NO would confirm this hypothesis.

Hence, in AMA pregnant women, we found statistical significance regarding group effect in the measurements of plasma TBARS, GSH and NO between AMA patients and controls.

Data about NO are of relevance for the physiologic regulation of pregnancy and for the onset of pregnancy-related pathologies. In the placental–umbilical unit, the NOS isoforms, namely the endothelial (eNOS) and the inducible ones, which are involved in NO release, are expressed in the syncytiotrophoblast (STB) and in the endothelium of umbilical vein and arteries and in placental microvascular endothelial cells [33]. NO, through its dilating effects, plays a pivotal role for maintaining endothelial function and is responsible for keeping vascular homeostasis, which ensures constant uterine blood flow [49]. Also, NO is engaged in STB endovascular invasion and development of the placenta through its angiogenic and vasculogenic effects [50].

For those reasons, changes in NO release, which can be observed in conditions of placental hypoperfusion and oxidative stress, may be involved in the onset of pregnancy-related diseases. In fact, in the presence of hypoxia and oxidative stress, the bioavailability of tetrahydrobiopterin, which is a cofactor of eNOS, is reduced and eNOS uncoupling can be observed. In this way, NO released from eNOS turns into peroxinitrites, which can reduce placental blood flow through vasoconstriction and activate inflammatory-response-associated signaling pathways [51].

In order to better understand the pathophysiology of pregnancy in AMA, both in relation to the effects on the mother and on the fetus, we performed some in vitro experiments on hUMSCs, which could play a fundamental role for the proper functioning of the placenta and the continuation of a pregnancy without complications [13]. The stimulation of those cells with plasma taken from AMA pregnant women was able to reduce cell viability and NO release and increase ROS release. It is noteworthy that the effects of plasma of AMA patients on hUMSCs persisted even after delivery.

It is also of note that a significant interaction group × time was found about NO and cell viability. This means that changes in response over time differed among AMA patients and controls. In particular, we observed an increase in cell viability of hUMSCs treated with plasma of controls, whereas no changes over time were observed for AMA patients.

Our data may have pathophysiological implications, since it is well known that hUMSCs can regulate trophoblast invasion, angiogenesis and placental function through the modulation of inflammation and the release of mediators like vascular endothelial growth factor, the placental growth factor, miRNAs or NO [16,21,52,53,54]. Alterations in the production of the aforementioned factors, with particular reference to NO, would, therefore, result in the loss of the role played by hUMSCs in the physiological regulation of pregnancy. For those reasons, our results, which highlight a possible mechanism underlying AMA pregnancy physiopathology, which could be related to unknown harmful circulating factors targeting hUMSCs, would add knowledge about this issue.

Regarding the nature of those “unknown circulating factors”, among others, we could speculate that extracellular vesicles (EVs) could be good candidates, since they are released by STB into the maternal circulation in increasing amounts with advancing gestational age to target various organs, and may represent a mechanism by which the placenta orchestrates maternal responses in normal and pathological pregnancy [54]. Actually, the aim of this study was not to investigate the nature of those circulating factors, which could be the object of subsequent research.

Overall, we could hypothesize that, in AMA pregnant women, a condition of feto-placental dysfunction related to aging phenomena would cause an imbalance between oxidants/antioxidants in favor of the former, with an increase in lipid peroxidation, which would worsen endothelial dysfunction. Furthermore, circulating factors released by the endothelium itself, the immune system and the uteroplacental tissues and cells, including hUMSCs, could have detrimental effects on maternal physiology, pregnancy progression, and fetal development. Thus, we could speculate that the oxidative imbalance we found in our study and the release of those factors could contribute to miscarriages, fetal developmental abnormalities, preterm birth, low birth weight and the development of cardiovascular or other chronic disease, which could arise in AMA pregnant women and in children too. However, those issues will need to be explored in greater depth in order to define their role in AMA pregnancy and AMA-related diseases.

Thus, it would be crucial to explore this field of research also from a prevention perspective and for giving recommendation to AMA pregnant women to decrease the AMA pregnancy-associated risks. In this regard, it could be suggested to change lifestyle in order to increase the antioxidant power and keep endothelial function.

Anyway, for the development of effective interventions aimed at mitigating the detrimental effects of oxidative stress during pregnancy, there should be an advance in the knowledge of the underlying mechanisms.

### Limitations

The results of this study could be implemented by increasing the sample size of both AMA pregnant women and controls as well and by carrying out analyses of plasma parameters even some months after delivery.

Although our study has possible relevant clinical implications, it should be, certainly, considered that the pathological conditions under examination could have many possible correlates, which have been investigated here and are not entirely well known.

Also, in our study, we used hUMSc from a depository since they are widely adopted to perform studies about inflammation, cancer and neurologic diseases [54,55,56]. In addition, the results we obtained were similar to those we found from preliminary experiments performed in primary hUMSCs isolated from umbilical cords. Anyway, the use of primary hUMSCs could be suggested for future studies about pregnancy and pregnancy-related diseases in order to better correlate the results obtained with the clinical ones and to increase the knowledge about their possible involvement in the physiopathology of AMA. In this regard, it could be interesting to perform crosstalk experiments between hUMSCs and other placental cell lines and analyze the role of EVs released by hUMSCs. As reported above, the nature of the “unknown circulating factors” should also be investigated as well by focusing on EVs.

Also, future analyses performed on hUMSCs through qPCR could be organized in order to investigate in detail the role of oxidants and antioxidants and of various NOS isoforms. Moreover, by means of confocal microscopy, it could be possible to implement knowledge on the cellular effects of the plasma of AMA pregnant women.

## 4. Materials and Methods

### 4.1. Patients

This observational case–control study was performed on 100 pregnant women older than 40 years (patients) and 20 pregnant women younger than 40 years (controls) enrolled at the Gynecology and Obstetrics Unit, Università del Piemonte Orientale, Azienda Ospedaliera Universitaria Maggiore della Carità, Novara, between 30 June 2021 and 30 June 2023, for a total of two years.

### 4.2. Clinical Evaluation

Patients and controls were subjected to anamnestic investigation. In each pregnant woman, blood samples were collected for the analysis of oxidants/antioxidants markers, such as TBARS, GSH and NO, as specified below. The inclusion criteria were singular or twin pregnancies; age ≥ 40 years for patients/<40 years for the controls; written informed consent; acceptance to the follow-up and giving birth at Gynecologic and Obstetrics Unit, Azienda Ospedaliera Universitaria Maggiore della Carità.

Women were excluded in cases of legal interdiction; childbirth in different centers; age < 40 years for patients and ≥40 years for the controls; positivity for SARS-CoV-2, anti-HIV or anti-HCV.

Plasma samples were collected at T0, T1, T2 and T3. T0 was taken at 11–13 weeks of gestational age, during the first trimester screening, when the patients were counseled and recruited. T1 was taken at 20–22 weeks of gestational age, during the second trimester morphological ultrasound. T2 was at the time of delivery and umbilical cord blood was collected. Finally, T3 was taken before Cesarean section or within 48–72 h after vaginal delivery.

Of the 134 AMA patients that visited in the selected study period, 34 women were excluded, as shown in Figure 3.

Therefore, 100 patients and 20 controls were enrolled. At various time points, some patients were lost, as shown in Figure 3 and Figure 4. Thus, the evaluations regarding the plasma redox balance and NO were executed on 100 patients at T0, 80 patients at T1, 43 patients at T2 and 53 patients at T3; and 20 controls at T0, 18 controls at T1, 16 controls at T2 and 18 controls at T3.

### 4.3. Biological Sample Analysis

In each pregnant woman, 15 mL blood samples were taken at 9 am in fasting conditions by using BD Vacutainer tubes (sodium heparin or ethylenediaminetetraacetic acid as anticoagulant). Each sample was centrifuged for 15 min through a centrifuge (Eppendorf, mod. 5702 with rotor A-4-38, Milan, Italy) at 1600 rpm and 4 °C. The plasma was aliquoted into 5 tubes, which were stored at −80 °C at the Physiology laboratory, Università del Piemonte Orientale, until they were used for the quantification of TBARS, GSH and NO and to perform the in vitro experiments on hUMSCs. Plasma samples were handled in pseudonymized conditions and measurements were performed at least in triplicate.

### 4.4. Quantification of Plasma TBARS

Plasma TBARS were measured by using the TBARS assay Kit (Cayman Chemical, Ann Arbor, MI, USA), which evaluates the MDA release [23,24]. Briefly, 100 µL of each plasma sample was added to sodium dodecyl sulfate solution (100 µL) and Color Reagent (2 mL). Samples were boiled for 1 h and then put on ice for 10 min to stop the reaction. After centrifugation for 10 min at 1600 g at 4 °C, 150 μL of each sample was transferred to 96-well plates for malonyldialdeide detection through a spectrophotometer (VICTOR™ X Multilabel Plate Reader; Perkin Elmer; Waltham, MA, USA) at 530–540 nm excitation/emission. A standard curve with the MDA Standard was performed as a reference for TBARS in each sample (expressed as malonyldialdeide in µM).

### 4.5. Quantification of Plasma GSH

Plasma GSH levels were quantified through the Glutathione Assay Kit (Cayman Chemical) [23,24]. Briefly, each plasma sample was deproteinated through the addition of meta-phosphoric acid solution. After centrifugation (2000× *g* for 2 min), the supernatant of each sample was collected and 50 μL/mL of TEAM reagent was added. Thereafter, 50 μL of each sample was moved to 96-well plates, where the measurement of GSH was executed through a spectrophotometer (VICTOR™ X Multilabel Plate Reader; Perkin Elmer) at 405–414 nm excitation/emission. To perform an accurate GSH quantification (as µM), a standard curve was prepared as well by using the GSH Standard.

### 4.6. Quantification of Plasma NO

Plasma NO was quantified by the Griess assay (Promega Italia Srl, Milan, Italy) [23,25]. To achieve this, 5 mL plasma sample was deproteinated by adding 10 mL sulfosalicylic acid. The samples were then vortexed every 5 min and allowed to react for 30 min at room temperature. After centrifugation (10,000× *g* for 15 min), 50 μL of the supernatant was added to saline (1:2 dilution) for the subsequent analysis. The remaining microliters were used without dilution. In order to reduce nitrate to nitrite, the samples were passed through a copper–cadmium column of an autoanalyzer (Autoanalyzer; Technicon Instruments Corp., Tarrytown, NY, USA) and then they were mixed with an equal volume of Griess reagents. After 10 min, the absorbance was measured by a spectrometer (VICTOR™ X Multilabel Plate Reader; Perkin Elmer) at 570 nm. NO release was examined in comparison with a standard curve and expressed as nitrites (μM).

### 4.7. In Vitro Experiments

#### 4.7.1. Effects of Plasma on hUMSCs

##### Cell Culture

The hUMSCs (StemBioSys, San Antonio, TX, USA) were cultured in α-Minimum Essential Medium Eagle (VWR International srl, Milan, Italy) supplemented with 2.5% heparin-free human pooled platelet lysate, 1% Antibiotic-Antimycotic (VWR International srl), and 1% GlutaMax (Euroclone, Pero, Italy).

Experiments were carried out using the plasma of 10 AMA patients at the time points, T0, T1 and T3, and of 5 patients belonging to the controls at the same time points. For the in vitro experiments, we selected patients/controls characterized by a physiological pregnancy, without the onset of any complications, and for whom samples were available and adequate at the three chosen time points.

Untreated cells were also included in the analysis. In particular, we used specific Transwell inserts in order to analyze cell viability, ROS and NO release in hUMSCs treated with plasma samples, as previously performed. As depicted in Figure 5, 10% plasma samples calculated in relation to total volume of each insert were positioned in the apical surface of the insert itself for 3 h, while hUMSCs were plated in the basal one. After 3 h stimulation with plasma, the inserts were removed and various assays were performed, as described below. Experiments were executed in triplicate and repeated three times on different pools of hUMSCs.

##### Cell Viability

Cell viability of hUMSCs was investigated by using 1% 3-[4,5-dimethylthiazol-2-yl]-2,5-diphenyl tetrazolium bromide (MTT assay; Cayman Chemical), as previously performed [25,56]. Briefly, after the 10% plasma stimulation of hUMSCs (50,000 cells/well in 24-well plates), the media were removed and 200 µL of the MTT solution diluted in Dulbecco’s Modified Eagle Medium high glucose w/o phenol red, supplemented with 2 mM L-glutamine and 1% penicillin-streptomycin, was added to each well. After the addition of 100 µL of dimethyl sulfoxide (Sigma, Milan, Italy) to dissolve the formazan crystals, cell viability was determined by measuring the absorbance through a spectrophotometer (VICTOR™ X Multilabel Plate Reader; PerkinElmer; Waltham, MA, USA), with a wavelength of 570 nm. Cell viability was calculated by setting control cells (untreated cells) as 100%.

##### ROS Release

ROS release by hUMSCs was analyzed through the DCFDA assay, which evaluated the oxidation of 2,7-dichlorodihydrofluorescein diacetate into 2,7-dichlorodihydrofluorescein (Abcam, Cambridge, UK) [25,56]. To achieve this, 50,000 hUMSCs/well were plated in 24-Transwells plates in complete medium, as executed for MTT assay. After stimulation with 10% plasma for 3 h, the medium was removed and washing was performed with phosphate buffer saline, which was followed by staining with 10 µM 2,7-dichlorodihydrofluorescein diacetate for 20 min at 37 °C. The fluorescence intensity of 2,7-dichlorodihydrofluorescein was measured at 485 nm/530 nm excitation/emission by using a spectrophotometer (VICTOR™ X Multilabel Plate Reader; PerkinElmer). Results were expressed as 2,7-dichlorodihydrofluorescein fluorescence intensity, which was proportional to the amount of intracellular ROS.

##### NO Release

NO release by hUMSCs was quantified through the Griess assay (Promega) [25,56]. To achieve this, 50,000 hUMSCs were plated in 24-Transwells plates in complete medium and then the same experimental protocol followed for MTT and ROS assays was used. At the end of the stimulations, NO was examined in the sample’s supernatants by adding Griess reagent following the manufacturer’s instruction. Thereafter, the absorbance of each sample was read at 570 nm through a spectrometer (VICTOR™ X Multilabel Plate Reader; Perkin Elmer). A standard curve was prepared to quantify the NO production, which was expressed as nitrites (μM).

##### Statistical Analyses

All data were collected using the Research Electronic Data Capture software, v10.3.3 (RED-Cap, Vanderbilt University, Nashville, TN, USA). The mean of the multiple measurements taken on each patient/control was considered for the analysis. For plasma quantification and the in vitro experiments, the results were presented as median and interquartile (IQR) range. The differences in quantitative variables between the two groups were assessed using the Mann–Whitney test. Spearman’s correlation coefficient was used to calculate the correlation between quantitative variables. The association between categorical variables was assessed using the chi-square test. Laboratory data were analyzed through repeated-measures ANOVA. This method of analysis was used because of the multiple responses taken in sequence on each experimental unit. The objective was to examine and compare response trends over time, comparing groups averaged over time and comparing measurement time within a group. The unstructured covariance matrix was used as a correlation pattern among responses to the same subject. A *p*-value < 0.05 was considered statistically significant. Statistical analysis was performed by using STATA version 18 (College Station, TX, USA: Stata Corp LLC), SAS 9.4 (SAS Institute Inc., Cary, NC, USA) and Graph PAD (GraphPad Prism 6 Software, San Diego, CA, USA).

## 5. Conclusions

Our results have highlighted an altered redox balance and a reduction in NO in plasma of AMA pregnant women and evidenced the existence of unknown circulating factors capable of inducing damages in hUMSCs. Those alterations were shown early and persisted up to 48 h after delivery. Beyond the possible obstetric complications, the increased plasma levels of TBARS, the reduced plasma levels of GSH and the harmful circulating factors could represent risk factors for the development of cardiovascular or other chronic disease, which could arise in women after advanced maternal age pregnancy and in children [13].

The results of this study about the role of oxidative stress in AMA pregnant women could have clinical implications regarding the usefulness of preventive or therapeutic treatments aimed at contrasting it.

Considering the role of the altered redox balance in the physiopathology of AMA pregnancy and its possible deleterious systemic effects, our research could correlate with the advice to act above all on lifestyle. In particular, a healthy and varied diet and moderate physical activity appear to be fundamental elements for the modulation of the redox state and the keeping of the mothers’ and children’s health.

Moreover, in the future, our data could have interesting clinical repercussions in terms of diagnosis, treatment and follow-up of several widespread diseases related to AMA pregnancy.

## Figures and Tables

**Figure 1 ijms-25-04869-f001:**
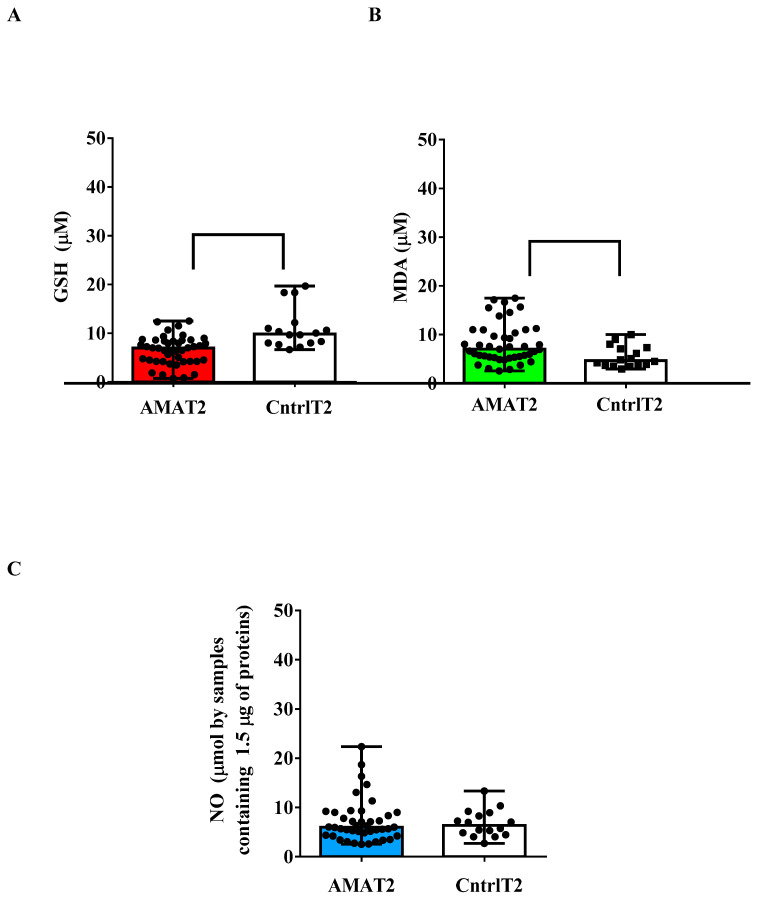
Glutathione (GSH, (**A**)), malonyldialdeide (MDA, (**B**)) and nitric oxide (NO, (**C**)) in umbilical cord plasma older (AMA > 40 yrs) and younger (Cntrl < 40 yrs) pregnant women collected at delivery. MDA was measured through the TBARS assay. T2: at the delivery. The results are expressed as median and range of different measurements. Square brackets indicate significance between groups (*p* < 0.05).

**Figure 2 ijms-25-04869-f002:**
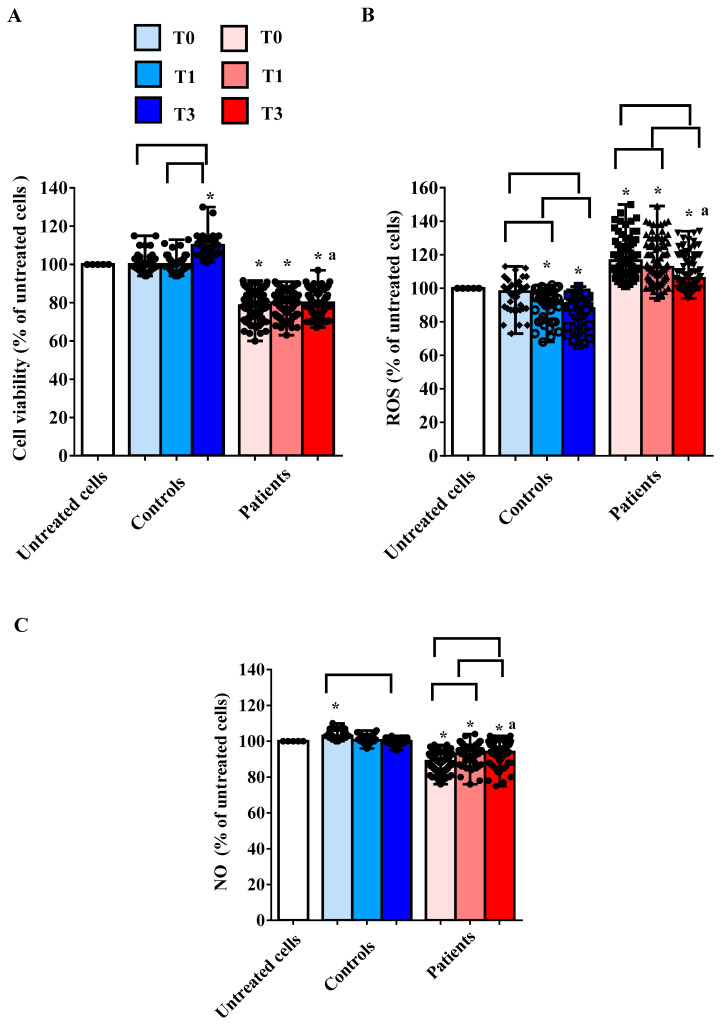
Effects of plasma taken from pregnant women older (patients) and younger than 40 years (controls) on cell viability (**A**), reactive oxygen species (ROS) release (**B**) and nitric oxide (NO) release (**C**) in hUMSCs. The bars represent the effects of plasma of all 10 patients and all 5 controls at various time points. T0: at 11–13 weeks of gestational age; T1: at 20–22 weeks of gestational age; T3: before Cesarean section or at 48–72 h after the delivery. The results are the median and range of repeated experiments. Untreated cells: non-treated hUMSCs. * *p* < 0.05 vs. untreated cells; a: *p* < 0.05 vs. T3 controls; square brackets indicate significance between the groups (*p* < 0.05).

**Figure 3 ijms-25-04869-f003:**
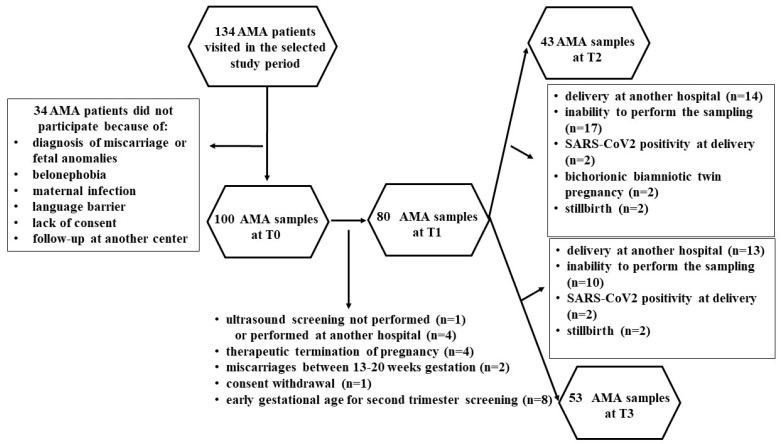
Flowchart describing advanced-maternal-age (AMA) pregnant women that underwent the analysis of plasma redox balance and NO at various time points.

**Figure 4 ijms-25-04869-f004:**
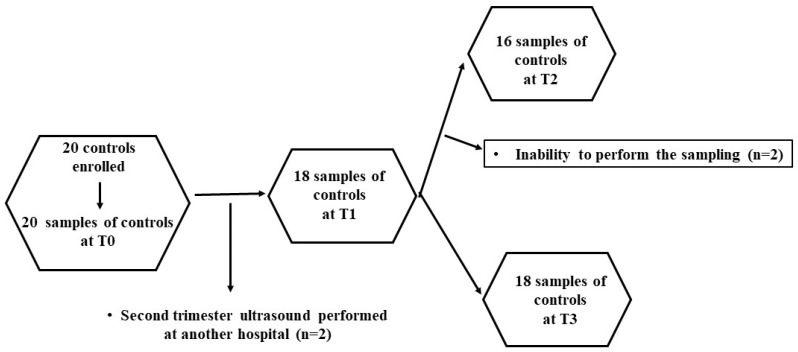
Flowchart describing the controls that underwent the analysis of plasma redox balance and NO at various time points.

**Figure 5 ijms-25-04869-f005:**
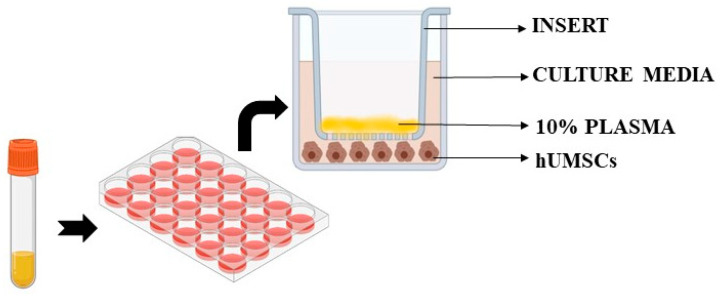
Experiments on hUMSCs. hUMSCs: human umbilical cord mesenchymal stem cells.

**Table 1 ijms-25-04869-t001:** Characteristics of the total population at enrollment.

	AMA Patients (*n* = 100)	Controls (*n* = 20)	*p*-Value
Age median (IQR)	41 (40–42)	30 (27–32)	<0.0001
BMI before pregnancy (kg/m^2^) median (IQR)	23.7 (20.9–27.4)	21.5 (19.8–23.3)	0.04
Ethnicity *n* (%)			
Caucasic	80 (80.0)	18 (0.90)	
Other	20 (20.0)	2 (0.10)	0.47
Nulliparous *n* (%)	36 (36.0)	7 (35.0)	0.93
Miscarriages *n* (%)			
0	45 (45.0)	12 (60.0)	
1	37 (37.0)	7 (35.0)	
>=2	18 (18.0)	1 (5.0)	0.29
Before-pregnancy pathologies *n* (%)	18 (18.0)	1 (5.0)	0.25
Positive combined prenatal screening test and positive at second-level test	26 (26.0)	1 (5.0)	0.06

**Table 2 ijms-25-04869-t002:** Delivery outcomes in patients and controls who gave birth by the end of the study.

	AMA Patients (*n* = 53)	Controls (*n* = 18)	*p*-Value
Type of delivery *n* (%)			
Spontaneous vaginal	23 (43.4)	11 (61.1)	0.19
Induced	5 (9.4)	3 (16.7)	0.40
Cesarean section	22 (41.5)	3 (16.7)	0.06
Dystocic	3 (5.7)	1 (5.6)	0.99
Gestational age at delivery median (IQR)	39 weeks (38–40)	39 weeks (38–40)	0.50
Birth weight at delivery (g) mean (SD)	3073 (575.2)	3288 (445.9)	0.15

**Table 3 ijms-25-04869-t003:** Response trends over time obtained by comparing groups averaged over time and measurement time within a group in plasma measurements.

	F Value	*p* Value	Outcomes
Group	42.45	<0.0001	TBARS
Time	0.40	0.67	
Interaction group × time	0.43	0.65	
Type of delivery	0.61	0.61	
Group	29.20	<0.0001	NO
Time	8.56	0.0004	
Interaction group × time	1.21	0.30	
Type of delivery	0.69	0.56	
Group	101.49	<0.0001	GSH
Time	3.89	0.02	
Interaction group × time	2.31	0.11	
Type of delivery	0.30	0.82	

GSH: glutathione; NO: nitric oxide; TBARS: thiobarbituric acid reactive substances.

**Table 4 ijms-25-04869-t004:** Response trends over time obtained by comparing groups averaged over time and measurement time within a group in the in vitro experiments.

	F Value	*p* Value	Outcomes
Group	53.03	<0.0001	NO
Time	1.87	0.20	
Interaction group × time	10.41	0.003	
Type of delivery	2.58	0.11	
Group	184.9	<0.0001	CELL VIABILITY
Time	9.34	0.004	
Interaction group × time	7.30	0.01	
Type of delivery	0.21	0.88	
Group	32.34	0.0001	ROS
Time	31.47	<0.0001	
Interaction group × time	0.45	0.65	
Type of delivery	1.33	0.31	

NO: nitric oxide; ROS: reactive oxygen species.

## Data Availability

The data that support the findings of the present study are available from the corresponding author upon reasonable request.

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
