# Peer review of "Plasma Redox Balance in Advanced-Maternal-Age Pregnant Women and Effects of Plasma on Umbilical Cord Mesenchymal Stem Cells"

_ijms, 2024, doi:10.3390/ijms25094869_

Round 1
Reviewer 1 Report
Comments and Suggestions for Authors
The manuscript submitted by Elena Grossini and co-authors to Int J Mol Sci is devoted to analyzing the possible influence of oxidative processes in the blood plasma of pregnant women of advanced age on some physiological and biochemical parameters.
The presented version of the manuscript requires improvements.
1) The article's text (Introduction, Discussion) does not explain why these parameters were chosen for analysis. For each of the parameters - TBARS, NO, GSH, and others, at least one paragraph should be added in the Introduction, with Refs.
2) The Discussion section should discuss which physiological meaning the results obtained from the analysis of biochemical parameters and hUMSCs may have
Author Response
The manuscript submitted by Elena Grossini and co-authors to Int J Mol Sci is devoted to analyzing the possible influence of oxidative processes in the blood plasma of pregnant women of advanced age on some physiological and biochemical parameters.
The presented version of the manuscript requires improvements.
1) The article's text (Introduction, Discussion) does not explain why these parameters were chosen for analysis. For each of the parameters - TBARS, NO, GSH, and others, at least one paragraph should be added in the Introduction, with Refs.
We have added some paragraphs in the Introduction (lines 54-57, 61, 62, 67-80, 93, 94, 99-101) and Discussion (265-267, 276-288, 314-328) about TBARS, GSH and NO and new references, as well.
2) The Discussion section should discuss which physiological meaning the results obtained from the analysis of biochemical parameters and hUMSCs may have
In the Discussion we have added more paragraphs about the physiologic meanings of our results (lines 286-288, 298-304, 363-379).

Reviewer 2 Report
Comments and Suggestions for Authors
1. It would be advantageous for the structure of the paper to adhere to a traditional format, commencing with an Introduction, followed by Materials and Methods, Results, Discussion, and concluding with Conclusions, as prescribed by the Journal's guidelines. If there is no intention to adjust the sequence, could you please provide a rationale for placing the Results section before the Methodology? Is this due to the length and complexity of the Methodology?
2. The authors should consider incorporating a section that outlines the study's limitations and proposes directions for future research.
3. In the Conclusions section, lines 414 and 415 refer to several widespread diseases associated with AMA pregnancy. Please enumerate examples of such diseases and, if possible, relate your findings to each disease/type of disease mentioned.
Otherwise the Results section provides a comprehensive view of the subject and data visualisation plots are well chosen.
Comments on the Quality of English LanguageThe quality of English can be improved.
Author Response
- It would be advantageous for the structure of the paper to adhere to a traditional format, commencing with an Introduction, followed by Materials and Methods, Results, Discussion, and concluding with Conclusions, as prescribed by the Journal's guidelines. If there is no intention to adjust the sequence, could you please provide a rationale for placing the Results section before the Methodology? Is this due to the length and complexity of the Methodology?
We followed the format, which can be found in papers published in IJMS. See https://www.mdpi.com/1422-0067/25/7/4122
- The authors should consider incorporating a section that outlines the study's limitations and proposes directions for future research.
We have added a section about limitations and possible proposals for future research (lines 381-402).
- In the Conclusions section, lines 414 and 415 refer to several widespread diseases associated with AMA pregnancy. Please enumerate examples of such diseases and, if possible, relate your findings to each disease/type of disease mentioned.
We have added the required information now (lines 232-249).

Reviewer 3 Report
Comments and Suggestions for Authors
The manuscript entitled "Plasma redox balance in advanced maternal age pregnant 2 women and effects of plasma on umbilical cord mesenchymal 3 stem cells" covers very important area. Authors analyzed women with advanced maternal age and their serum in comparison to younger women. The main drawback of the current manuscript is that authors did not provide description (or at least ideas) of detailed mechanisms which can affect on the fetus or can lead to miscarriage or some other anomalies in the case of AMA.
Below I will attach my certain questions and recommendations:
1. Line 76.
"Thus, our primary endpoint was to examine the plasma levels of lipid markers of peroxidation, GSH and NO at the first trimester (T0), at the second trimester during the morphologic ultrasound (T1), and at 48-72 hours after the delivery (T2), in pregnant women over 40 years compared with pregnant women younger than 40 years."
It is important to point here the median age of analyzed women. Younger that 40 can also be 18 or 22.
2. Table 2. Calculate p-values for all types of delivery.
3. Provide in Introduction some information about TBARS and actuality of their investigation.
4. Figures 1 and 2. P and C, why did you selected these abbreviations? In is better to use AMA instead P and Cntrl instead C.
5. In Discussion you write "which could be related to unknown harmful circulating factors targetting hUMSCs". Provide some ideas which can be these unknown factors.
6. Line 305 "TBARS, GSH e NO" - y NO.
7.
4.8.1.1. Why did you used cells from depository and not the cells obtained from the umbilical cord of babies of that patients which plasma you used in the experiments? Discuss how it can influence on the experiment results.
8.
4.8.1.2. It will be good to provide images from fluorescent of confocal microscope of stained for live/dead cells if possible.
9. Also if it is possible you may perform qPCR analysis for hUMSCs treated by serum for some genes involved in ROS and NOS answer.
10. Basing on your findings discuss which you could recommend for AMA-pregnant women to decrease the risks.
Author Response
The manuscript entitled "Plasma redox balance in advanced maternal age pregnant 2 women and effects of plasma on umbilical cord mesenchymal 3 stem cells" covers very important area. Authors analyzed women with advanced maternal age and their serum in comparison to younger women. The main drawback of the current manuscript is that authors did not provide description (or at least ideas) of detailed mechanisms which can affect on the fetus or can lead to miscarriage or some other anomalies in the case of AMA.
Below I will attach my certain questions and recommendations:
- Line 76.
"Thus, our primary endpoint was to examine the plasma levels of lipid markers of peroxidation, GSH and NO at the first trimester (T0), at the second trimester during the morphologic ultrasound (T1), and at 48-72 hours after the delivery (T2), in pregnant women over 40 years compared with pregnant women younger than 40 years."
It is important to point here the median age of analyzed women. Younger that 40 can also be 18 or 22.
We have added the median ages, as requested, now (lines 98, 99).
- Table 2. Calculate p-values for all types of delivery.
We have added p values in Table 2.
- Provide in Introduction some information about TBARS and actuality of their investigation.
We have added more sentences about TBARS (lines 67-73, 99, 100).
- Figures 1 and 2. P and C, why did you selected these abbreviations? In is better to use AMA instead P and Cntrl instead C.
We have performed required changes in Figs 1 and 2.
- In Discussion you write "which could be related to unknown harmful circulating factors targetting hUMSCs". Provide some ideas which can be these unknown factors.
We have added a paragraph in Discussion about the possible nature of unknown circulating factors, now (lines 355-362).
- Line 305 "TBARS, GSH e NO" - y NO.
We have corrected the mistake
- 4.8.1.1. Why did you used cells from depository and not the cells obtained from the umbilical cord of babies of that patients which plasma you used in the experiments? Discuss how it can influence on the experiment results.
We agree with the Reviewer about the raised point. However, due to logistical reasons related to the correct procurement of the umbilical cords, we couldn’t be able to isolate the hUMSCs. We just did some preliminary experiments on cell viability by using them and the results obtained were similar to those obtained with the hUMSCs from depository. So, the use of these cells didn’t introduce any bias. In addition, these cells are widely used to perform studies about inflammation, cancer and neurologic diseases. Of course, we believe that the use of primary hUMSCs could be suggested for future studies about pregnancy and pregnancy-related diseases in order to better correlate the results obtained with the clinical ones and to increase the knowledge about their possible involvement in the physiopathology of AMA. We have added some paragraphs about this issue in Limitations (lines 388-394).
- 4.8.1.2. It will be good to provide images from fluorescent of confocal microscope of stained for live/dead cells if possible.
- Also if it is possible you may perform qPCR analysis for hUMSCs treated by serum for some genes involved in ROS and NOS answer.
Points 8.48.1.2 and 9.
We agree with the Reviewer but we don’t have any images from confocal microscope. It could be done in next phases of this study on primary hUMSCs. Also, the results we obtained could be further implemented in next studies by performing qPQR analyses of antioxidants/oxidants, NOS isoforms, possibly on primary hUMSCs. It would be also interesting to analyze the nature of unknown circulating factors, perform crosstalk experiments between hUMSCs and other placental cell lines and deepen the intracellular pathways.
We have added some sentences about those issues in Limitations (lines 395-402) and Discussion (lines 359-362).
- Basing on your findings discuss which you could recommend for AMA-pregnant women to decrease the risks.
We have some sentences about this issue in Discussion (lines 373-376) and Conclusion (lines 562-569).

Reviewer 4 Report
Comments and Suggestions for Authors The biggest concern with this manuscript is the statistics. They are presenting results of a treatment by time interaction without any overall p value. There is a clear treatment and time independent variables. Figures 1--there seems to be a treatment difference at least with some variables, but a treatment by time interaction seems unlikely, and without treatment by time interaction these post hoc comparisons are not valid. There needs to be consistency with TBARS vs MDA. MDA (as listed on the graph) is an indication of TBARS--but this is not consistent or clear. Line 125 refers to TBARS measurement and Figure 2B, yet Figure 2B is MDA. Figures 3, 4 and 5 is showing data incorrectly analyzed. Your statistical n is the patient--eventhough they took the plasma and treated cells in replicate--treatment is at the patient. Similar to control values and panel B--treatment should also be summarized and a clear treatment by time interaction is necessary for post hoc comparisons. Line 191. "damage" is an overstatement. Suggest revising. Line 208 - 209. Results were not found in the umbilical cords. Suggest revising "Similar results were noted in the present experiment using plasma from AMA patients." Line 225 - 228. This statement is an overstatement of the current research. I feel the current data is weak in the "prevention of the development of cardiovascular disease". Revise to reflect the current research conclusions. (ie. there is some increase in oxidative stress). Line 274 and 284. It is not clear that "twin" and "delivery" status was included in the statistical analysis. If it was, it should be clarified as being included as co-variants, if it was not, it needs to be. Line 317. Is the standard curve MDA? If so it should be listed as such.
Author Response
The biggest concern with this manuscript is the statistics. They are presenting results of a treatment by time interaction without any overall p value. There is a clear treatment and time independent variables. Figures 1--there seems to be a treatment difference at least with some variables, but a treatment by time interaction seems unlikely, and without treatment by time interaction these post hoc comparisons are not valid. There needs to be consistency with TBARS vs MDA. MDA (as listed on the graph) is an indication of TBARS--but this is not consistent or clear. Line 125 refers to TBARS measurement and Figure 2B, yet Figure 2B is MDA. Figures 3, 4 and 5 is showing data incorrectly analyzed. Your statistical n is the patient--eventhough they took the plasma and treated cells in replicate--treatment is at the patient. Similar to control values and panel B--treatment should also be summarized and a clear treatment by time interaction is necessary for post hoc comparisons. Line 191. "damage" is an overstatement. Suggest revising. Line 208 - 209. Results were not found in the umbilical cords. Suggest revising "Similar results were noted in the present experiment using plasma from AMA patients." Line 225 - 228. This statement is an overstatement of the current research. I feel the current data is weak in the "prevention of the development of cardiovascular disease". Revise to reflect the current research conclusions. (ie. there is some increase in oxidative stress). Line 274 and 284. It is not clear that "twin" and "delivery" status was included in the statistical analysis. If it was, it should be clarified as being included as co-variants, if it was not, it needs to be. Line 317. Is the standard curve MDA? If so it should be listed as such.
Our study does not evaluate the effect of any treatment in AMA pregnant women or controls. The measurements carried out were conducted in the same subjects at various times during pregnancy up to postpartum, as described in Materials (lines 421-426), and in Results too (lines 119, 120).
In Figure 1, it is shown that plasma levels of MDA (measured as TBARS, as reported in text), are higher in AMA pregnant women than controls at T0 (11-13 weeks), T1 (11-22 weeks), and T3 (within 48-72 hours after vaginal delivery). In the same time, GSH and NO levels are lower in AMA pregnant women than controls, at T0, T1 and T3.
Also, in AMA pregnant women, GSH levels increase at T3, and are higher than those found at T2, but remain lower than those found in controls at the same timing (Fig. 1B). The levels of NO in AMA pregnant women increase at T1 vs T0, but remain lower than those found in controls (Fig. 1 C).
In controls, GSH levels increase from T0 to T3 (Fig. 1 B) and NO levels increase from T0 to T1 and then decrease at T3 (Fig. 1 C).
As regarding TBARS, as specified in text (lines 99, 100, 122, 123, Figure legends 1 and 2, 265-267, 276-278), it is a widely adopted assay to measure MDA levels which refers to lipid peroxidation. We hope to have clarified the issue, now.
Figure 3, 4, 5. We agree with the Reviewer and have added more sentences about statistical analysis about response trends over time obtained by comparing groups averaged over time and measurement time within a group. Models were adjusted for type of delivery. We did not do the same for twin, since only one AMA patient and one control had twin, and performing adjusting for this variable was meaningless. The results of those analysis are described in Table 3.
In order to avoid any misunderstanding and clarify the issues, we have deleted panel A from Figures 3-5.
Line 191. "damage" is an overstatement.
We have revised and put “we showed that unknown factors capable of affecting viability and ROS release in hUMSCs”.
Line 208 - 209. Results were not found in the umbilical cords. Suggest revising "Similar results were noted in the present experiment using plasma from AMA patients."
We have performed required changes.
Line 225 - 228. This statement is an overstatement of the current research. I feel the current data is weak in the "prevention of the development of cardiovascular disease". Revise to reflect the current research conclusions. (ie. there is some increase in oxidative stress).
We have changed the paragraph, now: “In particular, our results could be helpful to increase knowledge about the role of oxidative stress in the physiopathology of AMA pregnancy and of systemic diseases, which could arise in AMA pregnant women [43, 44]”.
Line 274 and 284. It is not clear that "twin" and "delivery" status was included in the statistical analysis. If it was, it should be clarified as being included as co-variants, if it was not, it needs to be.
As above reported, models were adjusted for type of delivery. We did not do the same for twin, since only one AMA patient and one control had twin, and performing adjusting for this variable was meaningless.
Line 317. Is the standard curve MDA? If so it should be listed as such.
It’s MDA standard.

Round 2
Reviewer 1 Report
Comments and Suggestions for Authors
The text of the article does not clearly show what the reader can understand from the added Table 1.
The discussion of the results presented in Table 1 should be expanded in the text of the article (in the Results section). Also, the reviewer suggests the table should be reorganized since the meaning of the lines “type of delivery, group, time, Interaction group × time” in its present form is not clear
Author Response
We have revised the manuscript according to Reviewer's suggestions.
1. We added more paragraph in order to describe Table 1 (lines 106-117).
2. Also, we revised the text taking into account the data shown in Table 3 (lines 139-146; lines 319-323) and Table 4 (lines 182-186; lines 345-348).

Reviewer 3 Report
Comments and Suggestions for Authors
Authors performed all required corrections.
Author Response
We thank the Reviewer for his positive opinion
Reviewer 4 Report
Comments and Suggestions for Authors
Suggested changes are on the attached file.

English use is acceptable in this manuscript.
Author Response
We have performed required changes throughout the text.
All changes are in red.
We split Table 3 into two tables (Table 3 and 4) reporting results of plasma measurements and the in vitro experiments. Also, we revised the Results section (lines 139-146 and lines 182-186) and Discussion section (lines 319-323 and lines 345-348) in order to add comments about the results of Table 3 and 4.
In order to avoid any misunderstanding, we deleted Figure 1
We added a description of Table 1 and 2 in Results (lines 106-117; lines 124-131).
We modified Figure 1 and changed Figure 2 (we added the symbol for statistical significance between patients and controls at T3).
We specified which were the complications, lines 208-210.
We modified the reference about the risk of stillbirth and complications (lines 232-237)
We modified the text in Discussion, following the Reviewer’s suggestions (lines 272-288; lines 292-293; lines 298-302)).
As regarding EVs, we added more paragraphs in order to answer to one point raised by another Reviewer. Anyway, we modified the text, now (lines 358-359; lines 362-364).
We changed the text in Discussion as suggested by the Reviewer (line 371, 375, 376).

Round 3
Reviewer 1 Report
Comments and Suggestions for Authors
After revision and correction of the reviewers' comments, the article may be recommended for publication in IJMS